# Optical Genome Mapping Enables Detection and Accurate Sizing of *RFC1* Repeat Expansions

**DOI:** 10.3390/biom13101546

**Published:** 2023-10-19

**Authors:** Stefano Facchini, Natalia Dominik, Arianna Manini, Stephanie Efthymiou, Riccardo Currò, Bianca Rugginini, Elisa Vegezzi, Ilaria Quartesan, Benedetta Perrone, Shahedah Koya Kutty, Valentina Galassi Deforie, Ricardo P. Schnekenberg, Elena Abati, Anna Pichiecchio, Enza Maria Valente, Cristina Tassorelli, Mary M. Reilly, Henry Houlden, Enrico Bugiardini, Andrea Cortese

**Affiliations:** 1IRCCS Mondino Foundation, 27100 Pavia, Italy; elisa.vegezzi@mondino.it (E.V.); ilaria.quartesan01@universitadipavia.it (I.Q.); anna.pichiecchio@mondino.it (A.P.); enzamaria.valente@unipv.it (E.M.V.); cristina.tassorelli@unipv.it (C.T.); 2Department of Neuromuscular Diseases, University College London, London WC1N 3BG, UK; n.dominik@ucl.ac.uk (N.D.); a.manini@ucl.ac.uk (A.M.); s.efthymiou@ucl.ac.uk (S.E.); r.curro@ucl.ac.uk (R.C.); bianca.rugginini01@universitadipavia.it (B.R.); b.perrone@ucl.ac.uk (B.P.); valentina.deforie.19@ucl.ac.uk (V.G.D.); r.schnekenberg@ucl.ac.uk (R.P.S.); elena.abati@unimi.it (E.A.); m.reilly@ucl.ac.uk (M.M.R.); h.houlden@ucl.ac.uk (H.H.); e.bugiardini@ucl.ac.uk (E.B.); 3Department of Pathophysiology and Transplantation, University of Milan, 20122 Milan, Italy; 4Department of Neurology and Laboratory of Neuroscience, IRCCS Istituto Auxologico Italiano, 20149 Milan, Italy; 5Department of Brain and Behavioral Sciences, University of Pavia, 27100 Pavia, Italy; 6Department of Internal Medicine, Kulliyah of Medicine, International Islamic University Malaysia (IIUM), Pahang 53100, Malaysia; shahedah@iium.edu.my; 7Department of Molecular Medicine, University of Pavia, 27100 Pavia, Italy

**Keywords:** optical genome mapping, Southern Blotting, CANVAS, RFC1, bionano, repeat expansion

## Abstract

A recessive Short Tandem Repeat expansion in *RFC1* has been found to be associated with cerebellar ataxia, neuropathy and vestibular areflexia syndrome (CANVAS), and to be a frequent cause of late onset ataxia and sensory neuropathy. The usual procedure for sizing these expansions is based on Southern Blotting (SB), a time-consuming and a relatively imprecise technique. In this paper, we compare SB with Optical Genome Mapping (OGM), a method for detecting Structural Variants (SVs) based on the measurement of distances between fluorescently labelled probes, for the diagnosis of *RFC1* CANVAS and disease spectrum. The two methods are applied to 17 CANVAS patients’ blood samples and resulting sizes compared, showing a good agreement. Further, long-read sequencing is used for two patients to investigate the agreement of sizes with either SB or OGM. Our study concludes that OGM represents a viable alternative to SB, allowing for a simpler technique, a more precise sizing of the expansion and ability to expand analysis of SV in the entire genome as opposed to SB which is a locus specific method.

## 1. Introduction

Cerebellar ataxia, neuropathy and vestibular areflexia syndrome (CANVAS) is caused by recessive short tandem repeat expansions in the second intronic region of *RFC1* [1,2,3]. Subsequent studies have confirmed that biallelic *RFC1* repeat expansions were shown to represent a common cause of late onset ataxia and sensory neuropathy, explaining 22% and 34% of genetically unconfirmed cases, respectively [1,4].

The underlying genetic cause of CANVAS is quite complex. Although most patients carry biallelic AAGGG expansions, additional pathogenic repeats were identified including ACAGG in Asians and AGGGC in Europeans [3,5,6,7]. Notably, all pathogenic repeats, independently from exact repeat motif, are large, ranging from around 250 to over 4000 repeats, while non-pathogenic expansions are typically < 100 repeats. Furthermore, although this has not been demonstrated yet in *RFC1* CANVAS, in other repeat expansions disorders, the length of the repeat expansions has an important prognostic role as larger expansions typically lead to an earlier onset and more severe phenotype. Biallelic *RFC1* repeat expansions were shown to represent a common cause of late onset ataxia and sensory neuropathy, explaining 22% and 34% of genetically unconfirmed cases, respectively [1]. Therefore, detection of the presence and measurement of the size of *RFC1* repeat expansion is of paramount diagnostic importance. Moreover, a small number of patients with typical CANVAS has been found to carry one AAGGG expanded allele and a second, truncating variant, in trans, warranting additional investigations in heterozygous carriers with suggestive clinical features [8,9,10].

Current diagnostic strategy for *RFC1* testing relies on polymerase chain reaction (PCR), including flanking PCR and a repeat-primed PCR (RP-PCR) [1]. However, given the large size and high GC content of the pathogenic AAGGG motif, PCR-based techniques fail to amplify the full expanded repeat. Therefore, demonstration of the presence of two expanded alleles and measurement of their size was only possible with traditional Southern Blotting (SB). SB utilises a pre-designed probe that only binds to a specific locus flanking the *RFC1* repeat, and the expansion sizing is based on the visual comparison between the sample track and a reference ladder track [1]. Despite being clinically very useful, SB is a time-consuming technique which requires considerable amount of work and a dedicated laboratory setup.

Southern Blotting has been a gold standard technique for measuring allele sizes in various conditions such as *C9orf72* repeat expansion disorders [11], myotonic dystrophy type 1 (DM1) [12], and fragile X syndrome [13]. SB is a cumbersome method and studies have been carried out whether more convenient and high-throughput methods, so far mainly limited to PCR, can replace or minimise the need for SB [14,15]. However, PCR cannot amplify large repetitive sequences; therefore, it is not possible to use it for sizing of repeat expansions. 

Optical Genome Mapping (OGM) is a new technology which enables accurate detection of large (>500 nucleotides) Structural Variants, based on the measurement of the distance between fluorophore-labelled probes which tag ultra-high molecular weight DNA molecules. The advantages of this technique include the following: (1) a more streamlined laboratory protocol; (2) the possibility of mapping the entire genome for each sample, instead of a single locus; (3) the possibility of automatizing the data analysis. The main commercial implementation of OGM is currently provided by Bionano Genomics, which is the technology used for this paper. Bionano OGM was able to reliably detect the presence of repeat expansion in DM1 and SCA10 [16,17]. In addition, OGM was successfully used to confirm the presence of biallelic *RFC1* expansions in seven Dutch patients carrying *RFC1* expansions [18]. However, a systematic comparison between OGM and SB was never performed.

In the present study, we compared OGM with SB for the detection and sizing of *RFC1* repeat expansion and showed that OGM can be a viable high-throughput alternative to SB for *RFC1* expansion testing, as part of its ability to assess the presence of structural variants and large repeat expansion at genome-wide level. 

## 2. Materials and Methods

### 2.1. Samples

A total of 17 CANVAS patients carrying biallelic AAGGG (n = 15) or ACAGG (n = 1) or compound heterozygous AAGGG and AGGGC (n = 1) repeat expansions were enrolled at the Institute of Neurology (IoN), University College London, Mondino IRCCS Foundation and the International Islamic University of Malaysia. DNA was extracted from fresh blood collected in EDTA blood tubes. All samples were subjected to SB and OGM and 2 samples (patients 7 and 10) were sequenced using long-read sequencing. Two control samples without CANVAS disease were also included in the OGM analysis.

### 2.2. Southern Blotting

Five ug of genomic DNA were enzymatically digested with EcoRI for 3 h and subsequently electrophoresed on 1.2% agarose for 15 h. The gel was washed for 45 min in depurination, denaturation and neutralising solutions. Subsequently, the fragmented DNA was transferred to a positively charged membrane using upward transfer method for 15 h. The DNA was then UV cross-linked on the membrane, and it was hybridised with a mixture of salmon sperm and *RFC1*-specific probe in digoxigenin granules solution (DIG Easy Hyb™ Granules, Roche, Basel, Switzerland) overnight at 49 °C. The membrane was washed and blocked. Anti-DIG antibody was added and incubated for 30 min at room temperature. Band visualisation followed incubation with detection buffer and chemiluminescent CDP-STAR substrate. The repeat sizing was estimated against DIG-labelled DNA molecular weight marker II after subtracting the wild-type allele size. For each individual, the sizes of detected alleles were recorded as number of pentanucleotide repeats. 

Densitometric profiles for SB lanes were obtained with ImageJ v1.54f (https://imagej.nih.gov/ij/ accessed on 13 August 2023).

### 2.3. Optical Genome Mapping

Ultra-high molecular weight DNA was extracted using “Blood and cell culture DNA isolation kit” according to the manufacturer’s protocol. Following Qubit Fluorometer (Invitrogen, Waltham, MA, USA) quantification, 750 ng DNA per sample was labelled using Bionano Prep DLS Labelling Kit. All samples were loaded onto Saphyr chips for linearisation and imagining and processed on a Bionano Saphyr machine (Bionano Genomics, San Diego, CA, USA). Molecules were aligned to the hg38 reference using the align_mol_to_ref.py script available in Bionano Solve 3.6 software package (https://bionano.com/software-downloads/ accessed on 13 August 2023).

*RFC1* repeat expansion is located between markers 7723 and 7724 of chromosome 4 (hg38; chr4:39343732-39350590; reference intermarker distance = 6858 bp). 

Instead of relying on the single size estimate provided by the Bionano Access platform, we decided to use all the mapping information available by collecting the intermarker distances over all molecules overlapping both markers. A histogram of the distances reveals the presence of one or two alleles, visible as peaks in the plot. The distribution of distances is decomposed in Gaussian components, using the Gaussian Mixture model from Scikit-learn python package (https://scikit-learn.org/stable/ accessed on 13 August 2023). The mean of each component provides an estimate for the allele size. The standard deviation accounts for technical errors in the optical reading and possibly for somatic instability. For each individual, the sizes of detected alleles were recorded as number of pentanucleotide repeats. 

The method comprises two steps:For each sample, we apply the clustering DBSCAN algorithm from Scikit-learn, with epsilon parameter obtained using the KneeLocator method (https://github.com/arvkevi/kneed accessed on 13 August 2023): all observations not belonging to a cluster are removed as outliers;We fit two Gaussian Mixture Models to the remaining observations, one with a single component and one with two components. The two-component model is preferred if both these conditions are satisfied: (a) the Bayesian Information Content (BIC) improves; (b) both components have weights > 25%

We refer to Appendix A for an implementation of the algorithm.

### 2.4. Targeted RFC1 Long-Read Sequencing

Targeted long-read sequencing (LR) was performed on patient 10 with Oxford Nanopore technology, and on patient 7 with PacBio technology. Target enrichment was performed with clustered regularly interspaced short palindromic repeats CRISPR/CRISPR-associated protein-9 nuclease (Cas9) system.

High molecular weight DNA was extracted from blood using the Qiagen (Venlo, The Netherlands) MagAttract HMW DNA Kit, and quality control was performed with Thermo Scientific NanoDrop. Subsequently, Agilent Femto Pulse Genomic DNA 165 kb kit was used to distinguish samples with majority of fragments over 25 kb that were used for CRISPR/Cas9 targeted sequencing as previously described [3]. Briefly, libraries were prepared from 5 µg of input DNA and sequenced using Nakamura et al. [19] CRISPR-Cas9 guides RFC1-F1: 5′-GACAGTAACTGTACCACAATGGG-3′, RFC1-R1: 5′-CTATATTCGTGGAACTATCTTGG-3′, RFC1-F2: 5′-ACACTCTTTGAAGGAATAACAGG-3′ and RFC1-R2: 5′-TGAGGTATGAAT CATCCTGAGGG-3′ for patient 10 and guides RFC1-F3: 5′-GAAACTAAATAGAACCAGCC-3′ RFC1-R3: 5′-GACTATGGCTTACCTGAGTG-3′ for patient 7. Minimap2 was used for alignment of sequences to the hg38 reference.

### 2.5. Correlation

The statistical analysis of correlation between SB and OGM results was conducted with the standard Python package “statsmodel”. A linear regression model with constant was fitted using the Ordinary Least Square method (OLS) from the same Python package.

## 3. Results

### 3.1. Clinical and Demographic Features

Demographic and clinical features of the patients included in our analysis (n = 17) are presented in Table 1. Except for one patient coming from Malaysia, the remaining cohort were of Caucasian origin, mainly from the United Kingdom and Italy. Median age at onset was 56 years (IQR, 39–61), and median disease duration at examination was 9 years (IQR, 4–15.5). Six patients showed all the three core clinical features of CANVAS, namely sensory neuropathy, cerebellar syndrome and vestibular dysfunction. Six patients had a complex neuropathy, including one case with sensory neuropathy and vestibular areflexia and five with sensory neuropathy and cerebellar ataxia, while five patients had an isolated sensory neuropathy. In three patients, the vestibular system was not examined. Thirteen patients exhibited chronic cough. Six patients showed dysautonomia, which was represented by erectile dysfunction in three of them (Pt 3, 5 and 6). One patient exhibited generalised, intermittent fasciculations (Pt 7). 

### 3.2. Technical Considerations of Southern Blotting and Optical Genome Mapping

SB relies on large quantities (5 µg) of high-quality and purity DNA. SB is compatible with most DNA extraction methods, thus facilitating sample processing and shipping of extracted DNA from collaborators across the globe. In comparison, OGM can only be performed on very high molecular weight DNA fragments (>150 Kbp), which requires a bespoke extraction method using the Bionano extraction kit from fresh or snap frozen blood or cell pellets. Hands-on processing time at the bench is 4 working days for SB and 2 working days for OGM, followed by Saphyr imaging and automatic data collection. 

SB size estimation relies on comparison to a ladder tract. OGM relies on fluorescent labels which bind to specific 6 bp DNA motifs (CTTAAG) present in the genome at an average of 20 times per 100 Kbp.

In addition to good technical skills, necessary for both methods, OGM requires computer literacy for size estimation in the online Bionano Access analysis platform, or to perform custom analysis.

Representative examples of OGM and SB are shown in Figure 1. 

### 3.3. Southern Blotting and Optical Genome Mapping Show Good Sizing Agreement

All CANVAS samples were confirmed to carry biallelic *RFC1* repeat expansions with both methods (see Table 2). Control samples were subjected to OGM analysis, confirming the absence of biallelic expansions (Control 1 has one expanded allele; Control 2 has two unexpanded alleles).

We observed an excellent linear correlation between the two methods (Figure 2), with r^2^ = 0.97. However, the linear coefficient is 0.62 [0.58–0.66] at 95% C.I., and the intercept is 232 [181–226] at 95% C.I. 

We show in Figure 3A the OGM molecules size distribution for all samples, with the estimated Gaussian components. We did not detect evidence of significant somatic instability, as suggested by a standard deviation of ~5% of the repeat size in all tested samples, independently from the repeat length. In Figure 3B, we show the SB images for all patients.

### 3.4. Targeted Long-Read Sequencing

We also performed a targeted long-read sequencing of the *RFC1* locus, as described in Methods, for two patients (Pt 7 and Pt 10).

For Pt 7, we obtained 2 PacBio CCS reads with expansion of 1160 and 1224 repeats; median = 1192.

For Pt 10, we obtained 20 Nanopore reads for the allele AGGGC, with expansions ranging from 233 to 5049 repeats; median = 3363 (IQR 3261–3532). No reads were obtained for the AAGGG allele.

Notably, the size of the expanded alleles as measured using LRS matches more closely the OGM data, while Southern Blotting tended to overestimate the size of large expanded alleles by 14–29% (Table 3).

## 4. Discussion

Biallelic *RFC1* expansions represent a common cause of late-onset ataxia and sensory neuropathy [1,2,3,4,5,6]. Unfortunately, it is still difficult to implement a diagnostic test. Indeed, the molecular diagnosis of *RFC1* CANVAS is challenging and currently relies on the combination of flanking PCR, repeat-primed PCR for different pathogenic (AAGGG, ACAGG, AGGGC, AAGGC and AGAGG) and non-pathogenic (AAAAG) repeat motifs, and SB as confirmation test, which is often not available in diagnostic labs. Therefore, the diagnostic implementation of *RFC1* genetic testing remains challenging. 

In this paper, we validated the OGM technology on 17 blood samples from patients carrying biallelic *RFC1* repeat expansions. We compared the repeat sizing between SB and OGM and showed a very good linear correlation of the two techniques. We noticed a deviation from the expected identity function in the regression, which is accounted by a systematic error either in the SB or in the OGM method, particularly for the expanded alleles over ~1000 repeats. This could either be due to overestimation of repeat size with SB or underestimation with OGM. SB relies on gel electrophoresis to resolve large fragments of genomic DNA. Possible formation of secondary structures by the repeats, slowing down the migration during electrophoresis, could lead to an overestimation of the repeat lengths. Moreover, due to the necessity of a visual comparison with a logarithmic scale, estimation of the allele size is increasingly imprecise for larger fragments, and it often cannot resolve similarly sized alleles resulting in a single band.

On the other hand, OGM may underestimate expansion size by taking into account kinked DNA molecules during imaging, leading to the underestimation of expansion size. 

To better assess the relative accuracy of the two methods, we compared the sizing estimates of SB and OGM with the size measured using targeted long-read sequencing of the expansion for Pt 7 and Pt 10. In both cases, we found a much better agreement with the OGM estimate (see Table 3). These results suggest that the discrepancy between OGM and SB is probably accounted for by a systematic error in the SB, possibly due to electrophoretic or analytic procedure as discussed above.

Moreover, OGM, unlike SB, was able to distinguish two alleles of similar size in 3 out of the 17 patients (Pt 6, Pt 9, Pt 17; see Table 2 and Figure 1B,C), while in one case (Pt 15), the presence of two distinct alleles was suggested via SB, but only one component was detected with OGM. Overall, OGM improved the allele sizing resolution in 4/17 (24%) samples. 

In addition, we showed that OGM is able to detect heterozygous carriers of RFC1 repeat expansions. This is of clinical relevance since detection of heterozygous expansion in patients with typical CANVAS symptoms should prompt additional testing including full sequencing of *RFC1* gene looking for a second nonsense variant [3,4].

An additional advantage of OGM is the possibility to screen for SV as well as large expansions (>500 nt) in the entire patient’s genome in parallel to *RFC1* testing. 

Both techniques require good technical skills, specific laboratory setups and special sample storage and transport considerations. However, advantages of OGM include a short response time (in ideal conditions, approximately 10 h hands-on time for DNA isolation and DNA labelling, overnight homogenisation of ultra-high molecular DNA, 8 h of run time at 100X coverage and 24 h for automated data collection), higher accuracy and high-throughput output. 

As part of this study, we have developed a bespoke algorithm to accurately capture all the available information on *RFC1* repeat size data and their dispersion compared to the standard DeNovo variant calling available on the Bionano Access software (currently at version 3.7), which only provide one value for each expanded allele. In particular, the in-house algorithm, based on the analysis of the Gaussian distribution of all available molecules spanning the repeat expansion, enabled accurate sizing of the *RFC1* repeat, as well as the assessment of its somatic instability, as indicated by the mean and the standard deviation of the Gaussian, respectively. Notably, based on the data generated in this study, OGM analysis did not support the presence of significant somatic instability of *RFC1* repeat in blood.

A known limitation of both OGM and SB is that they do not provide any information on the repeat sequence and need to be complemented with PCR, short or long-read sequencing. This is particularly true in cases with typical CANVAS symptoms but only heterozygous expansion where a truncating variant could be present in trans with the expansion, or in cases with suspected configuration motifs different to canonical pathogenic AAGGG.

In conclusion, OGM appears as a valid alternative to SB for the detection and sizing of *RFC1* expansions, along with genome-wide assessment of structural variants and other large repeat expansions, which could support its use in a diagnostic setting.

## Figures and Tables

**Figure 1 biomolecules-13-01546-f001:**
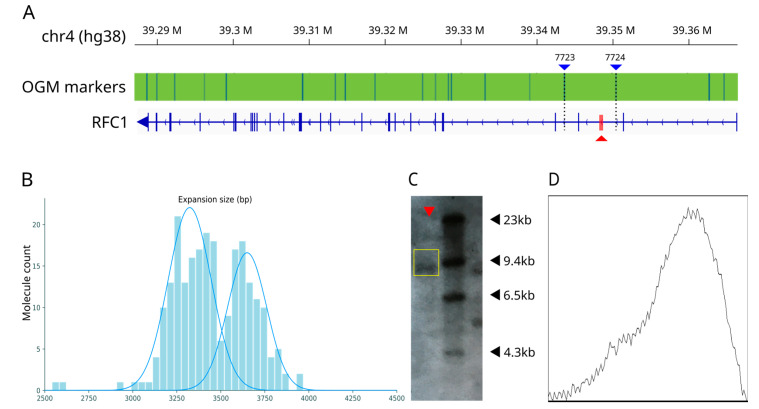
(**A**) Bionano OGM markers used for the analysis. The red triangle area indicates the position of the repeat expansion inside the second intron of *RFC1* (blue arrows point in the coding direction of the gene). The blue triangles indicate the position of the markers flanking the repeat (markers 7723 and 7724) (**B**) Optical genome mapping for Pt 6. Two alleles are observed as Gaussian components of size 664 and 730 repeats (3322bp and 3648bp, respectively) (**C**) Representative example of Southern Blotting plot. For Pt 6 (indicated by the red triangle), only one band is visible, corresponding to an expansion of 917 repeats (4585 bp) (original images can be found in Appendix A). (**D**) Densitometric profile for the SB band. Only one peak is visible.

**Figure 2 biomolecules-13-01546-f002:**
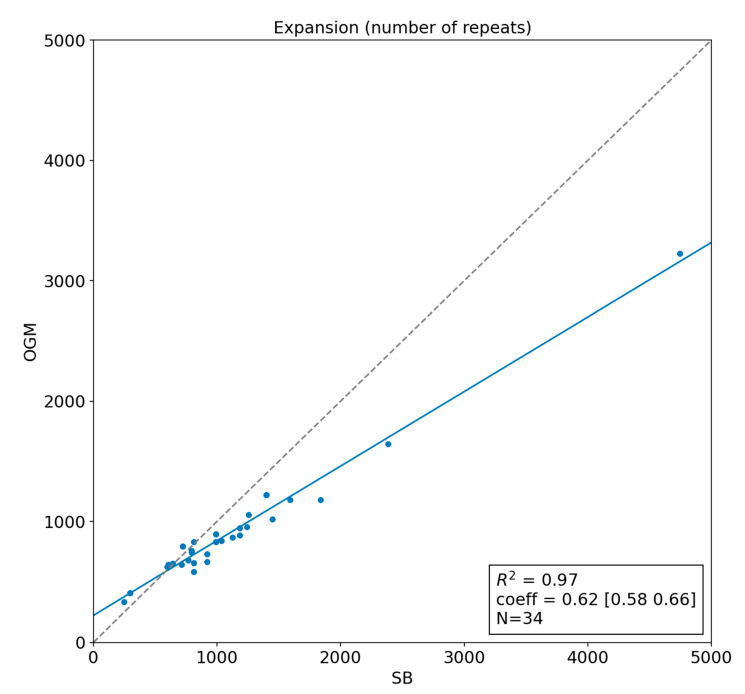
Linear regression showing the correlation between Optical Genome Mapping (OGM) and Southern Blotting (SB) size estimates. The grey dashed line represents the expected identity function in case of perfect correlation.

**Figure 3 biomolecules-13-01546-f003:**
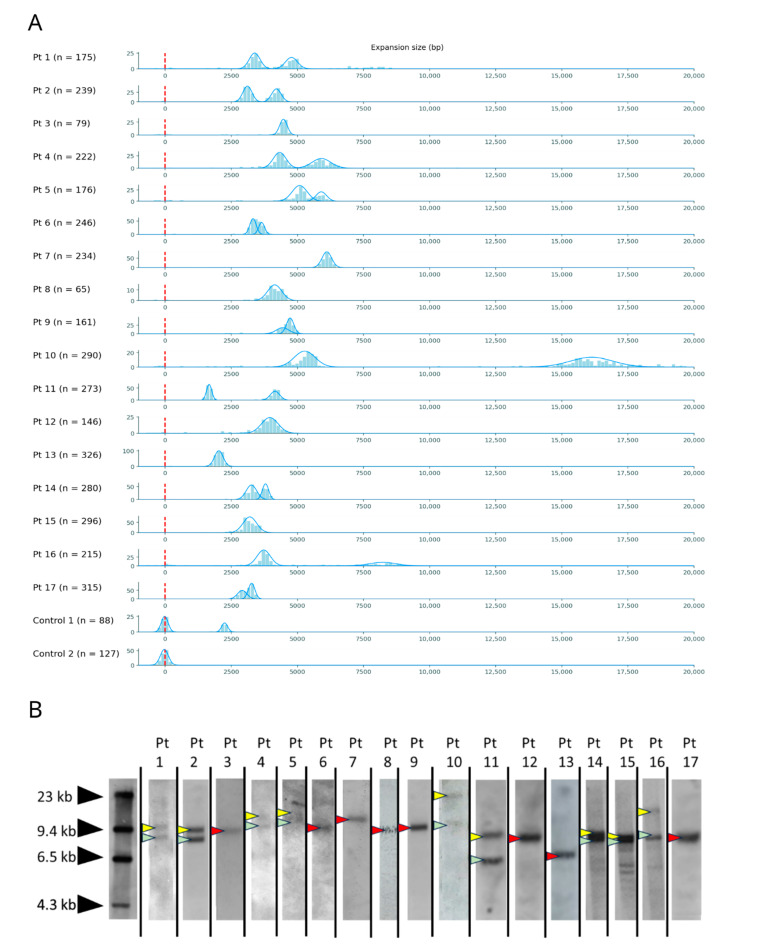
(**A**) OGM molecule size distribution for all samples, with estimated Gaussian components. On the vertical axis the molecule count is reported. The vertical dotted red line corresponds to a non-expanded allele. For each sample, we report the total number of observed molecules in parenthesis. (**B**) SB images for all patients. Arrows point to the alleles visible on Southern Blot; yellow and green when two alleles of distinct sizes are seen and red when two alleles of the same size are seen (original images can be found in Appendix A).

**Table 1 biomolecules-13-01546-t001:** Demographic and clinical features of the cohort. Pt: patient; F: female; M: male; AAO: age at onset; Y: yes; N: no; NA: not available.

Patient	Sex	Ethnicity	AAO	Disease Duration at Examination (Years)	Phenotype	Chronic Cough	Cerebellar Syndrome	Sensory Neuropathy	Bilateral Vestibular Areflexia	Dysautonomia	Use of Walking Aids (Age)	Additional Features
Pt 1	F	Caucasian (British)	39	4	Isolated sensory neuropathy	Y	N	Y	NA	N	No	/
Pt 2	F	Caucasian (British)	76	2	Complex neuropathy	Y	Y	Y	NA	N	Stick (78)	/
Pt 3	M	Caucasian (British)	35	43	CANVAS	Y	Y	Y	Y	Y	Stick (75)	Erectile dysfunction
Pt 4	F	Caucasian (British)	57	20	Complex neuropathy	Y	Y	Y	NA	N	Stick (74); Wheelchair (77)	/
Pt 5	M	Caucasian (British)	35	17	Isolated sensory neuropathy	Y	N	Y	N	Y	No	Erectile dysfunction
Pt 6	M	Caucasian (Italian)	59	6	CANVAS	Y	Y	Y	Y	Y	Stick	Erectile dysfunction
Pt 7	M	Asian (Malaysian)	38	29	CANVAS	Y	Y	Y	Y	N	Stick (63); Wheelchair (64)	Generalised, intermittent fasciculations
Pt 8	F	Caucasian (British)	59	14	CANVAS	N	Y	Y	Y	Y	Stick (72)	/
Pt 9	F	Caucasian (British)	25	40	CANVAS	Y	Y	Y	Y	Y	No	Thoracic syrinx
Pt 10	M	Caucasian (British)	71	11	Complex neuropathy	Y	Y	Y	N	Y	Wheelchair (81)	
Pt 11	F	Caucasian (Italian)	74	3	Isolated sensory neuropathy	Y	N	Y	N	N	No	/
Pt 12	F	Caucasian (Italian)	75	2	CANVAS	Y	Y	Y	Y	N	Stick (76)	/
Pt 13	M	Caucasian (Italian)	61	9	Isolated sensory neuropathy	N	N	Y	N	N	No	Diabetes
Pt 14	M	Caucasian (Italian)	51	9	Complex neuropathy	N	Y	Y	N	N	No	/
Pt 15	M	Caucasian (Italian)	56	4	Isolated sensory neuropathy	N	N	Y	N	N	Stick (58)	/
Pt 16	M	Caucasian (Italian)	41	4	Complex neuropathy	Y	N	Y	Y	N	No	/
Pt 17	F	Caucasian (Italian)	50	1	Complex neuropathy	Y	Y	Y	N	N	No	/
Control 1	F	Caucasian (British)	/	/	/	/	/	/	/	/	/	/
Control 2	F	Caucasian (Italian)	/	/	/	/	/	/	/	/	/	/

**Table 2 biomolecules-13-01546-t002:** Estimated sizes of the repeat expansions (number of pentanucleotide repeats). In OGM, repeat size is indicated as mean ± standard deviation of the Gaussian. Highlighted in grey are the patients where OGM, unlike SB, could better discriminate the size of the two expanded alleles.

Patient	SB Allele 1	SB Allele 2	OGM Allele 1	OGM Allele 2
Pt 1	765	1242	677 ± 41	955 ± 45
Pt 2	598	1035	622 ± 34	841 ± 36
Pt 3	989 (Homozygous)		894 ± 29 (Homozygous)	
Pt 4	1127	1593	866 ± 48	1182 ± 70
Pt 5	1447	1838	1017 ± 57	1180 ±40
Pt 6	917 (Homozygous)		664 ± 24	730 ± 22
Pt 7	1400 (Homozygous)		1223 ± 36 (Homozygous)	
Pt 8	991 (Homozygous)		829 ± 53 (Homozygous)	
Pt 9	1185 (Homozygous)		880 ± 46	943 ± 29
Pt 10	1256	4746	1055 ± 79	3226 ± 163
Pt 11	249	810	333 ± 20	831 ± 35
Pt 12	724 (Homozygous)		792 ± 63 (Homozygous)	
Pt 13	294 (Homozygous)		406 ± 32 (Homozygous)	
Pt 14	640	794	652 ± 40	759 ± 24
Pt 15	605	714	640 ± 51 (Homozygous)	
Pt 16	794	2386	745 ± 51	1646 ± 97
Pt 17	810 (Homozygous)		582 ± 35	654 ± 24
Control 1	/		−4 ± 26	450 ± 22
Control 2	/		−6 ± 30 (Homozygous)	

**Table 3 biomolecules-13-01546-t003:** Comparison of SB and OGM size estimates with long-read sequencing (number of pentanucleotides repeats). Pt 7 is homozygous, while for Pt 10, we report the estimates for the largest allele (with motif AGGGC). In both cases, we notice a much better agreement of LR with OGM.

Patient	SB	OGM	LR	|LR-SB|/SB	|LR-OGM|/OGM
Pt 7	1400	1223	1192	14%	3%
Pt 10	4746	3226	3363	29%	4%

## Data Availability

The data presented in this study are available on request from the corresponding author.

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
