# Peer review of "Optical Genome Mapping Enables Detection and Accurate Sizing of RFC1 Repeat Expansions"

_biomolecules, 2023, doi:10.3390/biom13101546_

Round 1
Reviewer 1 Report
The study by Facchini and co-workers explores the diagnostic use of Bionano optical genome mapping in CANVAS disease, a recessive neurological disorder triggered by pentanucleotide repeat expansions in the RFC1 gene. The study attempts to evaluate the accuracy of repeat sizing by OGM and the practical use of OGM in comparison to Southern blotting, the current gold standard.
The study comes from a world leading group in the field of CANVAS disease. The work has considerable technical merit and diagnostic impact, potentially warranting publication in the journal 'Biomolecules'. The genuine accuracy/precision of OGM in RFC1 expansion sizing need however to be further clarified and the potential diagnostic usefulness of OGM discussed in more detail.
Major points
1. The author's data suggest that Bionano OGM is more accurate for the sizing of pathological expansions than Southern blot. In particular, it is shown that OGM can resolve two pathological RFC1 alleles of similar expansion size estimated to about 3322 and 3648 base pairs, respectively, in patient 6 (Fig. 1B). Unfortunately, the corresponding Southern blot (Fig. 1C) is problematic due to high background. To demonstrate the superior precision of expansion sizing by OGM, the authors have to analyze the SB of Pat. 6 by densitometry. Please add the densitometry profile to the Figure. Please also consider to omit the unlabeled lanes on the left part of the SB.
2. The study data indicate a substantial discrepancy between repeat size estimates by OGM and SB. In Pat. 6 for instance, allele size measured by OGM is about 25% lower than allele size measured by SB. The precise reasons for this discrepancy remain unclear. The paper states that Southern blot tends to overestimate the size of large expanded alleles (l. 191). Would this be due to secondary structures formed by the pentanucleotide repeats slowing their migration during gel electrophoresis ? Yet, other kinds of bias related to OGM are also conceivable. First, OGM may underestimate expansion size by taking into account kinked DNA molecules during imaging, leading to underestimation of expansion size. Has this been corrected? Second, OGM (or the algorithm used here) may be skewed towards the detection of short alleles. To clarify this important issue, data from the litterature (references) on the accuracy of SB should be provided and study data should be re-analyzed. The graph on Fig. 2 should display the x- and y-axis at equal scale; the diagonale (y = x) should be added and the correlation analysis should be better explicited in the Methods section and in the Figure Legend. This should help to clarify whether the discrepancy in expansion sizing is due to biais from SB, OGM or both.
3. Fig. 3 is highly instructive. Please add the number of individual alleles measured. The corresponding Southern blots should be included as a panel of Fig. 3. Please make sure that distinct blots are separated by vertical lines. Please also make sure that all blots are correcly aligned/rotated.
4. The new analysis software should be made publicly available.
5. The turnaround time for OGM is indicated as 10 hours or four working days. This seems underestimated given the times required for the isolation of high molecular weight DNA, the slow dissolution of large DNA molecules, the DNA labeling using DLE-1 and the Saphyr imaging.
6. It would be interesting to discuss in more detail the use of OGM versus long read sequencing using Nanopore or PacBio for CANVAS disease. The paper by Nakamura et al. (targeted long read seuqencing of RFC alleles is cited by the authors.
Minor points
Abstract
l. 21 frequent cause of late onset ataxia ...
l. 27 blood samples
l. 29. The abstract concludes that OGM allows to detect SV in the entire genome. This conclusiion should be revisited since the SV analysis used here restricted to a portion of the RFC gene rather than genome-wide.
Introduction
l.46 PCR based techniques fail to amplify the full expanded repeat Please provide ref specific for CANVAS (RFC) PCR
l. 49 SB utilises a pre-designed probe that only binds to a specific locus flanking the RFC1 repeat, and the expansion sizing is based on the visual comparison between the sample track and a reference ladder track. >>Please provide reference
Methods
l.84 digoxygenin granules solution (DIG) . granules ??? >> Please correct
Results
Hands-on processing time is 4 working days for SB and 2 working days for OGM.
>> Please correct. See point 4 above
Discussion
l.226 the allele sizing through SB may be imprecise
l.230 blood samples
l.238 OGM include a short response time (approximately 10 hours for DNA isolation and library preparation). >> conventional OGM using bionano does not comprise library preparation. Please correct.
Figures
Fig. 1A >> Please indicate orientation of the RFC gene
Reviewer 2 Report
This interesting and practical article shows that the optical genomic mapping (OGM) technique is able to reliably detect pathogenic biallelic expansions of the RFC1 gene, responsible for CANVAS syndrome, in a small but sufficient cohort of patients. The article is well written. The figures are very clear.
Major suggestion
1) It seems to me that the authors should include a negative control in their results (ie the results of OGM obtained from a sample not carrying a pathogenic expansion, or being heterozygous for a pathogenic and a non-pathogenic allele). This may be trivial, but it seems to me that a negative controle would be easy to obtain and would reinforce the scientific validity, of which I have absolutely no doubt.
Minor suggestions
2) Line 35. CANVAS can also be caused by RFC1 nonsense and frameshift variants. This should be made clear in the introduction, in my opinion, since the main scope of the article is the molecular diagnosis of CANVAS (PMID: 35883251 and several other articles since 2022).
3) Line 36. Repeat conformation heterogeneity has been shown for CANVAS (PMID: 35355059). While this is specified in the other section of the manuscript, it would be useful to also mention this in the introduction, and to discuss the implication for optical mapping diagnosis.
4) Line 38. I don't believe that a correlation between the size of expansions and the severity of the disease has been shown for CANVAS. On the contrary, in cohorts age at onset appears to be independent of expansion size (PMID: 33666721).
5) Line 44. PCR stands for Polymerase chain reaction.
6) Discussion. It would be useful to know whether OGM can detect heterozygous carriers of pathogenic expansions because CANVAS can be caused by the compound heterozygous association of a pathogenic expansion and a truncating variant.
Round 2
Reviewer 1 Report
The authors should be congratulated for their efforts on repeat sizing in CANVAS disease using OGM (Bionano technology). The revised version of the author's manuscript responds to all issues raised. A few typos need to be corrected at the proof stage :
independently
Gaussian, referring to Carl Gauss
Southern, referring to Edwin Mellor Southern
due to
hands-on
Minor editing of English language required
Reviewer 2 Report
The authors have responded adequately to each of my suggestions.